# DNA-PKcs Inhibition Sensitizes Human Chondrosarcoma Cells to Carbon Ion Irradiation via Cell Cycle Arrest and Telomere Capping Disruption

**DOI:** 10.3390/ijms25116179

**Published:** 2024-06-04

**Authors:** Birgit Lohberger, Sandra Barna, Dietmar Glänzer, Nicole Eck, Andreas Leithner, Dietmar Georg

**Affiliations:** 1Department of Orthopedics and Trauma, Medical University of Graz, Auenbruggerplatz 5-7, 8036 Graz, Austria; dietmar.glaenzer@medunigraz.at (D.G.); nicole.eck@medunigraz.at (N.E.); andreas.leithner@medunigraz.at (A.L.); 2Department of Radiation Oncology, Medical University of Vienna, Währinger Gürtel 18-20, 1090 Vienna, Austria; sandra.barna@meduniwien.ac.at (S.B.); dietmar.georg@meduniwien.ac.at (D.G.); 3MedAustron-Ion Therapy Center, Viktor-Kaplan Strasse 2, 2700 Wiener Neustadt, Austria

**Keywords:** carbon ion irradiation, chondrosarcoma, AZD7648, DNA-PKcs inhibitor, telomere length

## Abstract

In order to overcome the resistance to radiotherapy in human chondrosarcoma cells, the prevention from efficient DNA repair with a combined treatment with the DNA-dependent protein kinase catalytic subunit (DNA-PKcs) inhibitor AZD7648 was explored for carbon ion (C-ion) as well as reference photon (X-ray) irradiation (IR) using gene expression analysis, flow cytometry, protein phosphorylation, and telomere length shortening. Proliferation markers and cell cycle distribution changed significantly after combined treatment, revealing a prominent G_2_/M arrest. The expression of the G_2_/M checkpoint genes cyclin B, CDK1, and WEE1 was significantly reduced by IR alone and the combined treatment. While IR alone showed no effects, additional AZD7648 treatment resulted in a dose-dependent reduction in AKT phosphorylation and an increase in Chk2 phosphorylation. Twenty-four hours after IR, the key genes of DNA repair mechanisms were reduced by the combined treatment, which led to impaired DNA repair and increased radiosensitivity. A time-dependent shortening of telomere length was observed in both cell lines after combined treatment with AZD7648 and 8 Gy X-ray/C-ion IR. Our data suggest that the inhibition of DNA-PKcs may increase sensitivity to X-rays and C-ion IR by impairing its functional role in DNA repair mechanisms and telomere end protection.

## 1. Introduction

For patients diagnosed with chondrosarcoma, treatment options are limited. Complete surgical resection remains the gold standard for both primary and recurrent chondrosarcoma, while radiation therapy and chemotherapy are also available as treatment options [1,2]. However, chondrosarcoma has a limited response to chemotherapy and radiotherapy. This extensive resistance is due to several factors: poor vascular supply, a slow cell division rate, and a hyaline cartilage matrix that hinders access to the cells. In addition, overall survival and prognosis depend on the histological grade and tumor subtype of this rather heterogeneous group of locally aggressive and malignant bone tumors [3]. Because of their low radiosensitivity, high doses are advised in palliative situations, after incomplete resection, or for non-operable tumors in difficult anatomical locations. Particle therapy with protons or carbon ions (C-ions) offers better local control and increased patient survival rates compared to conventional photon beam therapy [4].

The advantages of ion beam therapy lie in the distinctive physical attributes, especially the localized high dose deposition at a precisely defined depth known as the Bragg peak, coupled with a rapid dose fall-off beyond that peak [5]. These features enable high-precision dose delivery, ensuring excellent coverage of the target volume while minimizing exposure to normal tissues. In contrast to protons, carbon ions (C-ions) exhibit an enhanced radiobiological effect due to their high ionization density, which makes them the best current radiation therapy option for radiation-resistant tumors. On the other hand, the availability of C-ion therapy is limited, with only four centers in Europe offering this treatment modality, and 13 centers on a global scale. The established indications for C-ion therapy include skull base tumors, chordomas, and non-operable chondrosarcomas [6,7,8].

In a previous investigation, we demonstrated that exposure to both X-ray and proton irradiation (IR) led to the diminished survival of chondrosarcoma cells, a cell cycle arrest at the G_2_/M phase, and a shift in cellular metabolism. Nevertheless, most of the observed DNA damage was repaired within 24 h after proton IR, and the metabolic phenotype was subsequently restored [9]. Therefore, inhibiting these highly efficient DNA repair mechanisms is the logical next step for improving radiosensitivity and potentially the success of radiotherapy. We are able to demonstrate that the combined treatment with the potent and selective ATP competitive inhibitor of ATR (ATRi) VE-821 enhances the radiation sensitivity of chondrosarcoma cells and notably inhibits effective DNA repair mechanisms in vitro [10]. C-ion IR is also a promising route for the inhibition of DNA repair mechanisms, as the ionization density of C-ions is increased compared to protons.

The DNA-dependent protein kinase (DNA-PK) plays a crucial role in the DNA damage response (DDR), particularly within the non-homologous end-joining pathway (NHEJ), where it detects and repairs DNA double-strand breaks (DSBs) [11]. The primary mechanism for repairing DSBs induced by ionizing radiation heavily relies on DNA-PK, a complex composed of the Ku70 and Ku80 heterodimers along with the DNA-PK catalytic subunit (DNA-PKcs). It orchestrates the NHEJ repair pathway by phosphorylating factors such as Artemis and XRCC4 [12,13], as well as the DNA damage marker γH2AX [14]. Furthermore, DNA-PK is involved in various other cellular functions, such as modulating chromatin structure, maintaining telomeres, and regulating transcriptional processes [15]. In the context of radiation oncology and to investigate a potential improvement in radiation sensitivity, understanding how human chondrosarcoma cells respond to the inhibition of the DNA-PKcs pathway is of particular interest. In this study, we assessed the cell biological and radiosensitizing impacts of the DNA-PKcs inhibitor AZD7648 on human chondrosarcoma cell lines. We investigated its effects in conjunction with both conventional photon (X-ray) IR and carbon ion (C-ion) therapy. We focused on several key aspects, including cell proliferation, cell cycle distribution, the expression and phosphorylation levels of DNA damage markers, and potential effects on telomere length. Telomeres are protective caps situated at the ends of chromosomes and play a crucial role in maintaining genomic stability [16,17]. If the combined treatment of C-ion IR with a DNA-PKcs inhibitor were able to shorten the telomeres, the radiation sensitivity of chondrosarcoma could be increased.

## 2. Results

### 2.1. The Processes of Cellular Growth Regulation

The dose–response relationship was analyzed after a pretreatment with 0.03 and 100 µM of the DNA-PKcs inhibitor 7,9-dihydro-7-methyl-2-[(7-methyl[1,2,4]triazolo[1,5-a]pyridin-6-yl)amino]-9-(tetrahydro-2H-pyran-4-yl)-8H-purin-8-one (AZD7648) (Figure 1A) with and without an additional IR (Figure 1B), which only led to a slight reduction in cell viability. For all further experiments, we decided on concentrations of 1, 3, and 10 µM for the pretreatment. To surmount the resistance to radiotherapy in human chondrosarcoma cells, we investigated the efficacy of inhibiting the cell growth and DNA repair of AZD7648 in conjunction with C-ion IR or conventional photon IR. To demonstrate the regulation of critical proliferation-related genes such as cMyc (*MYC*) and cyclin D1 (*CCND1*), as well as the cell survival marker survivin (*BIRC5)* following the combined treatment, we conducted RT-qPCR analysis using RNAs isolated 24 h after treatment. Specifically, we examined the cells treated with 3 µM AZD7648 alone, or in combination with either 8 Gy X-ray IR (Figure 1C) or 8 Gy C-ion IR (Figure 1D). The dark grey bars represent the SW-1353 and the light grey dotted bars represent the Cal78 chondrosarcoma cells.

IR with 8 Gy as well as the combined treatment led to a significant increase in the expression of cMyc (X-ray: 1.38 ± 0.4 (SW-1353); 1.47 ± 0.3 (Cal78); C-ions: 2.14 ± 0.6 (SW-1353); 1.81 ± 0.5 (Cal78)) and cyclin D1 (X-ray: 2.76 ± 0.8 (SW-1353); 2.89 ± 0.7 (Cal78); C-ions: 1.93 ± 0.7 (SW-1353); 1.34 ± 0.5 (Cal78)). Survivin, one of the most important genes regarding cell proliferation and resistance to ionizing radiation, was highly significantly downregulated. In this context, additional IR significantly augmented the effect of AZD7648, with C-ions demonstrating a notably stronger impact (X-ray: 0.42 ± 0.2 (SW-1353); 0.87 ± 0.2 (Cal78); C-ions: 0.23 ± 0.1 (SW-1353); 0.42 ± 0.1 (Cal78)). All the mean values and their standard deviations (SD) as well as their significances are listed in Appendix A.

### 2.2. The Additive Effect of Combined Treatment with the AZD7648 on Cell Cycle Distribution

Another crucial aspect of tumor biology involves the disruption of the cell cycle caused by therapeutic interventions. Flow cytometry analysis was conducted to assess the impact of the various modes of IR in combination with AZD7648 treatment on the cell cycle distribution of chondrosarcoma cultures when exposed to 8 Gy X-rays/C-ions. Non-irradiated (0 Gy) cells and cells subjected to radiation alone were included as controls. Table 1 displays all the values obtained from each of the five individual experiments, indicating the percentage of gated cells, along with their corresponding mean ± SD and statistical differences (n = 5). The graphical representations of the G_0_/G_1_, S, and G_2_/M values of both cell lines are shown in stacked bars (Figure 2A). The representative flow cytometry measurements are presented in Figure 2B. C-ion IR induced a notably substantial rise in the number of cells in the G_2_/M phase compared to control conditions. This increase was accompanied by a reduction in the number of cells in both the G_0_/G_1_ and S phases, suggesting a sustained arrest in the G_2_/M phase at the 24 h time point. The addition of AZD7648 treatment significantly augmented the G_2_/M arrest induced by C-ion IR, whereby the S-phase almost disappears completely during combined treatment with C-ions. The Cal78 cell line exhibited a notably greater shift in cell cycle phases towards the G_2_/M phase, particularly in response to the AZD7648 treatment.

The protein expression levels of the key regulators of the G_2_/M phase of the cell cycle, including cyclin B, cyclin-dependent kinase (CDK)1, and p53 (*TP53*), were examined through immunoblotting in both cell lines 24 h and 72 h after IR (Figure 3). The analysis was conducted under control conditions (ctrl 0 Gy), IR with 8 Gy X-rays (Figure 3A), IR with 8 Gy C-ions (Figure 3B), and combined treatment with 1–10 µM AZD7648. An increase in cyclin B1 and CDK1 protein expression was observed 24 h after IR. However, the combined treatment with AZD7648 showed a dose-dependent downregulation of both checkpoint proteins, which regulate the transition to the G_2_/M phase. This is consistent with the observations from the flow cytometry data. The increased p53 expression caused by IR was reduced in a dose-dependent manner by the treatment with AZD7648. Protein band quantifications (Δ ratio to ctrl 0 Gy; mean ± SD) are listed in Appendix A.

Furthermore, RNA was isolated 24 h after X-ray IR (Figure 3C), respectively, and C-ion IR (Figure 3D) and the relative gene expression were analyzed. The dark grey bars represent the SW-1353 and the light grey dotted bars represent the Cal78 chondrosarcoma cells. Both IR and the combined treatment led to a significant decrease in the expression of the G_2_/M checkpoint key genes cyclin B (X-ray: 0.34 ± 0.1 (SW-1353); 0.51 ± 0.1 (Cal78); C-ions: 0.53 ± 0.2 (SW-1353); 0.59 ± 0.1 (Cal78)), CDK1 (X-ray: 0.67 ± 0.1 (SW-1353); 1.23 ± 0.2 (Cal78); C-ions: 0.59 ± 0.2 (SW-1353); 1.18 ± 0.2 (Cal78)), and WEE1 (X-ray: 0.49 ± 0.1 (SW-1353); 0.74 ± 0.1 (Cal78); C-ions: 0.56 ± 0.2 (SW-1353); 0.86 ± 0.2 (Cal78)).

### 2.3. Combined Treatment with DNA-PKcs Inhibition and Particle IR Affected AKT and Chk2 Phosphorylation and the Expression of Protein Stability Marker and Energy Sensor

As protein phosphorylation occurs rapidly, the samples for AKT and Chk phosphorylation were isolated 1 h after the combined treatment with 3 µM AZD7648, with or without exposure to 8 Gy X-ray (Figure 4A) or C-ion IR (Figure 4B). At the protein level, it was shown that the increased phosphorylation of AKT caused by IR is reduced in a dose-dependent manner by the combined treatment with AZD7648. In contrast, the phosphorylation of Chk2 increased significantly after the combined treatment. Protein band quantifications (Δ ratio to ctrl 0 Gy; mean ± SD) are listed in Appendix A.

The relative gene expression of the protein stability marker HSP27—a signal partner of AKT—was highly significant downregulated (X-ray: 0.75 ± 0.2 (SW-1353); 0.59 ± 0.2 (Cal78); C-ions: 0.88 ± 0.2 (SW-1353); 0.54 ± 0.2 (Cal78)) in both cell lines primarily by the IR (Figure 4C,D); the additional DNA-PKcs inhibition caused only minor changes. The gene expression of the cellular energy sensor AMPK was increased in response to the treatment (X-ray: 1.75 ± 0.4 (SW-1353); 1.67 ± 0.4 (Cal78); C-ions: 2.22 ± 0.4 (SW-1353); 1.27 ± 0.3 (Cal78)). The dark grey bars represent the SW-1353 and the light grey dotted bars represent the Cal78 chondrosarcoma cells.

### 2.4. DNA Repair and MDM2-p53 in Response to Radiation and DNA-PKcs Inhibition

To examine the regulation of the XRCC4 DNA repair marker and the MDM2-p53 feedback loop genes following the combined treatment, we conducted an RT-qPCR analysis with RNAs isolated 24 h after 3 µM AZD7648 with and without 8 Gy X-ray/C-ion IR. The analysis of the gene expression for these genes is depicted in Figure 5. The gene expression of the DNA damage marker XRCC4 increased 24 h after 8 Gy X-ray/C-ion IR (X-ray: 1.56 ± 0.3 (SW-1353); 1.40 ± 0.1 (Cal78); C-ions: 1.70 ± 0.2 (SW-1353); 1.78 ± 0.4 (Cal78)). Especially after the combined treatment with AZD7648 and C-ion IR, these values decreased highly significantly to 0.85 ± 0.1 (SW-1353); 1.02 ± 0.1 (Cal78) (Figure 5B). The feedback loop between MDM2 and p53 is recognized as crucial for controlling the activity and levels of p53 in response to stress. MDM2 gene expression increased significantly after both IR and combined treatment (X-ray: 3.76 ± 1.1 (SW-1353); 1.58 ± 0.4 (Cal78); C-ions: 4.55 ± 1.8 (SW-1353); 1.37 ± 0.4 (Cal78)). 

### 2.5. DNA-PKcs Inhibition Reduced the NHEJ DNA Repair and Resulted in Abbreviated Telomere Lengths after IR

The protein expression levels of the NHEJ key regulators Ku70, Ku80, Artemis, DNA-Ligase IV, and the DNA damage marker γH2AX were examined through immunoblotting in both cell lines 1 h and 24 h after IR (Figure 6). The analysis was conducted under control conditions (ctrl 0 Gy), IR with 8 Gy X-ray (Figure 6A), IR with 8 Gy C-ion IR (Figure 6B), and combined treatment with 1–10 µM AZD7648. Both X-ray and C-ion IR activated the phosphorylation of DNA-PKcs highly efficiently, which was inhibited by all concentrations of AZD7648. As phosphorylation is a very rapid process, this effect is particularly visible after 1 h. Twenty-four hours after IR, the NHEJ components were reduced by the combined treatment, which led to impaired DNA repair and increased radiosensitivity. The combined treatment also maintained the phosphorylation of γH2AX. Differences between the two types of IR could not be observed. Protein band quantifications (Δ ratio to ctrl 0 Gy; mean ± SD) are listed in Appendix A.

To investigate the influence of IR and DNA-PKcs inhibition on telomere length, DNA was isolated from the chondrosarcoma cells on day 3, day 10, and day 20 after IR and telomere length was analyzed by subsequent qRT-PCR (Figure 7). The inhibition of DNA-PKcs alone did not cause any changes. IR and in particular the combined treatment with AZD7648 gradually decreased telomere length time dependently. In comparison over time, both cell lines showed a significant decrease in telomere length on day 30 after the combined treatment. This accelerated telomere loss could contribute to an increased radiosensitivity of the chondrosarcoma cells.

## 3. Discussion

Treating malignant bone tumors like chondrosarcomas with C-ion RT appears promising with regard to tumor control, overall survival, and risk profile of early and late toxicity [18]. Limiting factors in this area of research are the rarity of these tumors and the fact that only a few centers in the world currently use this technology. C-ion IR causes a lower toxicity in the healthy tissue while being effective in the tumor region due to the energy deposition of the Bragg peak. Compared to conventional X-ray therapy, however, C-ion therapy has a higher relative biological effectiveness in the tumor due to the elevated LET. In general, the LET of C-ions varies with beam energy and SOBP position and length. The more fields are combined during the clinical setting of chondrosarcoma treatment, the lower the maximum LET becomes while maintaining an overall higher LET in the target [19].

In prior research, we demonstrated that human chondrosarcoma cells exhibit a remarkably effective DNA repair mechanism shortly after exposure to proton ionizing radiation. Additionally, this process was accompanied by a restructuring of cellular metabolism [9]. For this reason, an improved understanding of the cellular processes and pathways involved is of outstanding importance for overcoming radioresistance. Due to the multifaceted role of DNA-PKcs in a variety of cellular responses to DNA damage, such as NHEJ repair, cell cycle control, senescence, and telomere length maintenance, targeting DNA-PKcs is considered a promising strategy to explore novel radiosensitizers. Fok et al. demonstrated that the highly selective DNA-PKcs inhibitor AZD7648 is an efficient sensitizer of radiation- and doxorubicin-induced DNA damage [20] and enhanced the therapeutic efficacy for patients with ovarian cancer [21]. The aim of the present study was to investigate the potential effect of DNA-PKcs inhibition on human chondrosarcoma cells. For this purpose, the cells were subjected to combined treatments with the DNA-PKcs AZD7648 and conventional photon IR on the one hand, and particle IR with C-ions on the other hand. In addition to the cell cycle distribution, the results from protein expression and protein phosphorylation, as well as gene expression analyses, provide us with insight into the altered cell biology.

The oncogenic transcription factor c-Myc serves as a master regulator of cellular growth and metabolism, and was identified as a radiosensitive locus in breast cancer [22]. cMyc activation may reduce long-term cell viability by inducing DNA damage and activating the senescence program [23]. Our data revealed a significant increase in cMyc (*MYC*) and cyclin D1 expression after both X-rax and C-ion IR in chondrosarcoma cells. In addition, the effect of AZD7846 is significantly increased in both cell lines with IR. Cyclin D1 (*CCDN1*) functions as a molecular link connecting cell cycle regulation, adhesion, invasion, and the interplay between tumor, stroma, and the immune system in cancer [24]. We suggest that the increased expression of cMyc and cyclin D1 promotes the development of DNA damage and related cell cycle changes. Survivin (*BIRC5*) serves as a significant cancer biomarker, granting tumor cells enhanced survival capabilities by suppressing apoptosis. An siRNA screening targeting 51 apoptosis-related genes in chondrosarcoma cells identified survivin as crucial for the survival of chondrosarcoma cells [25]. The highly significant decrease observed with X-ray and C-ion IR and especially with the combined AZD7648 treatment suggests a sensitization of the cells.

When DNA is damaged, a complex network of signaling cascades is activated to ensure the survival of the cell. These tightly regulated processes, such as cell cycle arrest, give the cell the time it needs to repair the lesion or initiate apoptosis and senescence [26]. Checkpoints are monitoring mechanisms that control the order and integrity of the major events occurring during the cell cycle. The primary drivers of cell cycle progression are the cyclin-dependent kinases (CDKs). Binding to cyclins allows inactive CDKs to adopt an active conformation similar to monomeric and active kinases [27]. Our data clearly showed that G_2_/M cell cycle arrest at an early time point after IR and combined treatment with AZD7648 also correlated well with the upregulation of cell cycle regulatory protein p53 expression and the downregulation of cyclin B1, CDK1, and WEE1 at the protein and gene expression level.

The AKT family of serine/threonine protein kinases plays a central role in governing various facets of cell functions, encompassing proliferation, viability, metabolic processes, and the onset of tumorigenesis. AKT is activated by DNA damage as well as growth factors [28]. In contrast, active AKT can promote DNA repair through NHEJ and inhibit checkpoint signaling and repair. After activation, ATM initiates the phosphorylation of various substrates at the site of damage, including γH2AX, Chk2, and the tumor suppressor p53, thereby orchestrating cell cycle arrest, DNA repair, or apoptosis and serves as an upstream activator of AKT after γ-radiation [29]. AKT phosphorylation was activated by both X-ray and C-ion IR and reduced in a dose-dependent manner by a combined treatment with AZD7648. This also indicates increasing radiosensitivity.

Additionally, it has been shown that AKT exists in a signaling module with HSP27 [30]. The HSP27 (*HSPB1*) gene encodes a member of the small heat shock protein family. Its presence has been linked to unfavorable clinical outcomes across various human cancers. This protein might facilitate cancer cell growth and spread, while also shielding them from programmed cell death [31]. We were able to show a highly significant downregulation after combined treatment with DNA-PKcs inhibition and IR. AMPK, a critical metabolic sensor, inhibits protein biosynthesis and protects cells from stresses [32,33]. The combined treatment of the chondrosarcoma cells with IR and AZD7648 led to a highly significant increase in expression.

The next gene of interest is *MDM2*, which plays a crucial role in cellular responses to both ionizing and UV radiation. The level of *MDM2* expression determines the extent to which radiation triggers an increase in the activity of the p53 tumor suppressor [34]. *MDM2* acts as a survival factor in several cell types by limiting the apoptotic function of p53 (*TP53*). In addition, *MDM2* expression is triggered in response to DNA damage, resulting in increased levels of the MDM2 protein. These heightened levels are believed to abbreviate the duration of the cell cycle arrest initiated by p53 in response to radiation. Our results have shown that X-ray IR and, to a greater extent, C-ion IR increased *MDM2* and p53 (*TP53*) expression, especially in SW-1353 chondrosarcoma cells. Additional DNA-PKcs inhibition with AZD7648 caused a further small increase. The fact that SW-1353 cells have a TP53 mutation could play a role in the clear difference between the two cell lines [35].

DNA-PKcs forms a tight complex with Artemis and is the only active protein kinase described in the NHEJ pathway [36]. DNA-PK is composed of the DNA-binding Ku70/80 heterodimer and the catalytic subunit DNA-PKcs. They assemble at DNA ends to form the active DNA-PK complex, which initiates DSB repair through NHEJ [37]. In protecting telomeres from fusion, both Ku and DNA-PKcs play a crucial role. In normal somatic cells, telomeres generally shorten progressively with each cell division. When the telomeres become critically short, the cells enter a state of replicative senescence, which causes the cells to stop dividing. However, cancer cells must overcome this limitation to allow uncontrolled growth and reach a state of immortality. Zhou et al. showed that MCF-7 and Hela cell lines with shorter telomeres were more susceptible to C-ion IR [38]. Telomere length is therefore a potential target for new cancer therapies. For this reason, we investigated whether C-ion IR and a combined treatment with the DNA-PKcs inhibitor AZD7648 can influence telomere length. Our present data suggest that the inhibition of DNA-PKcs by AZD7648 may increase cellular sensitivity to C-ion IR by disrupting its functional role in telomere end protection.

All data obtained in this study indicate that DNA-PKcs inhibition by AZD7648 increases the radiosensitivity of human chondrosarcoma cells and thus makes them more responsive to radiotherapy. Although X-ray and C-ion IR have different physical properties including their vast differences in ionization density, i.e., LET values, they revealed similar biological effects in cell culture experiments. This might be explained by the experimental IR conditions, which do not take into account fractionation effects and the micro-environment of the tumor. C-ion RT offers the option of LET optimization while maintaining a uniform dose in the target instead of relying exclusively on dose optimization, as in X-ray RT. This leads to lower recurrence [18,39]. Despite the comparable biological effects that we established in this manuscript, the advantages of PT for the patient, which result from the radiation–physical properties, must not be ignored.

## 4. Materials and Methods

### 4.1. Cell Culture 

The SW-1353 (ATCC^®^ HTB-94™, LGC Standards, Middlesex, UK) and Cal78 (ACC449; DSMZ, Leibniz, Germany) chondrosarcoma cell lines were cultured in Dulbecco’s modified Eagle’s medium (DMEM-HG) supplemented with 10% FBS, 1% L-glutamine, 1% penicillin/streptomycin, and 0.25 µg amphotericin B (all GIBCO^®^, Invitrogen, Darmstadt, Germany). These cell lines were authenticated by STR profiling within the last three years. All the experiments were performed with mycoplasma-free cells. For IR experiments, adherent chondrosarcoma cells in the log-growth phase were plated either in a density of 1 × 10^5^ cells/Slideflasks 9 cm^2^ (Thermo Fisher Scientific, Waltham, MA, USA) or 5 × 10^5^ cells/T25 flasks and incubated overnight at 37 °C with 5% CO_2_.

### 4.2. Experimental Irradiation Conditions

All the photon (X-ray) and C-ions experiments were performed at the Austrian ion therapy and research facility MedAustron (Wiener Neustadt, Austria). X-ray reference IR was conducted using a 200 kV beam (YXLON Y.TU 320-D03, YXLON GmbH, Hamburg, Germany) with a nominal dose rate of 1.3 Gy/min. For radiobiological experiments, the following filtration setup was employed: 3 mm Be, 3 mm Al, and 0.5 mm Cu. C-ion IR was performed in an experimental research room with a horizontal beam line, employing active spot scanning with energy variation. The biological samples were embedded in a water phantom for horizontal beams (PTW, Freiburg, Germany), which was positioned with a precision robot couch and a laser system indicating the isocenter. Respective treatment plans covering the sample size with a homogenous dose (field size of 17 × 9 cm^2^) were created with the treatment planning system (TPS) RayStation (RaySearch Laboratories, Stockholm, Sweden). The resulting treatment plan had a spread-out Bragg peak length (SOBP) of 4 cm (covering depths between 6 and 10 cm) based on energy layer spacings of either 1 mm or 2 mm and energies ranging from 170 to about 239 MeV/u. The ionization density along the SOBP, expressed as dose averaged linear energy transfer (LETd), was in the range between 13 and 206 keV/µm. Within the central region of the C-ion SOBP, where the cells were positioned, the impact of positioning errors remains minimal. Assuming a generous positioning uncertainty of ±0.5 mm results in a LETd of 56 ± 1 keV/μm. All LETd values were derived from Monte Carlo calculations performed with the Geant4-based toolkit GATE.

### 4.3. Viability and Proliferation Analysis

For the dose–response relationship, the chondrosarcoma cells were pretreated with 0.03–100 µM 7,9-dihydro-7-methyl-2-[(7-methyl[1,2,4]triazolo[1,5-a]pyridin-6-yl)amino]-9-(tetrahydro-2H-pyran-4-yl)-8H-purin-8-one (AZD7648; Selleckchem, Houston, TX, USA). Afterwards, the cells were irradiated with 0 Gy (non-IR control) or a total dose of 8 Gy. Cell viability was determined with the CellTiter-Glo^®^ cell viability assay (Promega Corporation, Madison, MI, USA) and normalized to the untreated controls. Background reference values were derived from the culture media. Absorbance was measured with a LUMIstar™ microplate luminometer (BMG Labtech GmbH, Ortenberg, Germany) (mean ± SD; n = 3, performed in biological quadruplicates).

### 4.4. Flow Cytometry Cell Cycle Analysis

Twenty-four hours after pretreatment with 3 µM AZD7648 (Selleckchem) and X-ray/C-ion IR with 0 Gy and 8 Gy, the cells were harvested by trypsinization and fixed with 70% ice-cold ethanol for 10 min at 4 °C. Before flow cytometry analysis, the cell pellet was resuspended in propidium iodide (PI)-staining buffer (50 μL/mL PI, RNAse A) and incubated for 15 min at 37 °C. Cell cycle distribution was measured with CytoFlexLX (Beckman Coulter, Pasadena, CA, USA) and analyzed using the ModFit LT software Version 4.1.7 (Verity software house). Five independent experiments were conducted in each case.

### 4.5. Protein Expression Analysis

Whole-cell protein extracts were prepared with a lysis buffer (50 mM Tris-HCl pH 7.4, 150 mM NaCl, 1 mM NaF, 1 mM EDTA, 1% NP-40, 1 mM Na3VO4) and a protease inhibitor cocktail (P8340; Sigma Aldrich, St. Louis, MI, USA), after 1 h, 24 h, and 72 h, respectively, after X-ray/C-ion IR. The following groups have been isolated: untreated non-IR controls (ctrl 0 Gy), untreated IR controls (ctrl 8 Gy), and 1 µM, 3 µM, and 10 µM AZD7648 treated irradiated samples. Protein concentration was determined with the Pierce BCA Protein Assay Kit (Thermo Fisher Scientific). The proteins were separated by SDS-PAGE and were blotted on Amersham™ Protran™ Premium 0.45 µM nitrocellulose membranes (GE Healthcare Life Science, Little Chalfont, UK). Primary antibodies against the cell cycle regulators cyclin B, cyclin-dependent kinase (CDK)1, p53; the serine/threonine kinases p-AKT/AKT and p-Chk2; and the key players of the NHEJ repair pathway p-DNA-PKcs/DNA-PKcs, Ku70/80, Artemis, DNA-Ligase IV, phospho-histone γH2AX; and β-actin (all Cell Signaling Technology, Danvers, MA, USA) as loading control were used. The blots were developed using a horseradish peroxidase-conjugated secondary antibody (Dako, Jena, Germany) for 1 h and the Amersham™ ECL™ prime Western blotting detection reagent (GE Healthcare). Chemiluminescence signals were detected with the ChemiDocTouch Imaging System (BioRad Laboratories Inc., Hercules, CA, USA) and images were processed with the ImageLab 5.2 Software (BioRad Laboratories Inc.).

### 4.6. Reverse Transcription Polymerase Chain Reaction (RT-PCR)

Total RNA was isolated 24 h after combined treatment with 3 µM AZD7648 and 8 Gy X-ray/C-ion IR using the RNeasy Mini Kit and DNase-I treatment according to the manufacturer’s manual (Qiagen, Hilden, Germany). Two µg RNA were reverse transcribed with the iScript-cDNA Synthesis Kit (BioRad Laboratories Inc.) using a blend of oligo(dT) and hexamer random primers. Amplification was performed with the SsoAdvanced Universal SYBR Green Supermix (Bio-Rad Laboratories Inc.) using technical triplicates and measured by the CFX96 Touch (BioRad Laboratories Inc.). The following QuantiTect primer assays (Qiagen) were used for real-time RT-PCR: cMyc, cyclin D1, and survivin (proliferation); cyclin B1, CDK1, and WEE1 (cell cycle regulation); XRCC4, MDM2, and p53 (DNA damage); heat shock protein (HSP)27 (protein stability); and AMP-activated protein kinase (AMPK) (energy sensor). The results were analyzed using the CFX manager software for CFX Real-Time PCR Instruments (Bio-Rad Laboratories Inc., version 3.1). The software and quantification cycle values were exported for statistical analysis. The results with Ct values greater than 32 were excluded from the analysis. The relative quantification of expression levels was obtained by the ∆∆Ct method based on the geometric mean of the internal controls ribosomal protein, large, P0 (*RPL*), and TATA box binding protein (*TBP*), respectively. The expression level (Ct) of the target gene was normalized to the reference genes (ΔCt), and the ΔCt of the test sample was normalized to the ΔCt of the control (ΔΔCt). Finally, the expression ratio was calculated with the 2^−ΔΔCt^ method (n = 6; biological triplicates).

### 4.7. Telomere Length Measurements

On days 3, 10, and 20 after the combined treatment with 3 µM AZD7648 and 8 Gy X-ray/C-ion IR total genomic DNA was isolated using the QIAamp^®^ DNA Mini Kit (Qiagen) according to the manufacturer’s protocol. The commercially available PCR kit from ScienCell Research Laboratories Inc. (San Diego, CA, USA) was utilized to determine the average relative telomere length of genomic DNA. In this process, telomere-specific primers identified and amplified telomeric sequences. For each DNA sample, two consecutive reactions were carried out: The first involved amplifying a single-copy reference (SCR) gene, targeting a 100 bp region on human chromosome 17, which served as a reference for calculating the telomere length of the target samples. The PCR reactions were conducted in a final volume of 20 μL, using 5 ng of reference/genomic DNA sample, 2 μL of telomere primer, and 10 μL of 2× Master Mix. The PCR conditions were as follows: 95 °C for 10 min, followed by 32 cycles of 95 °C for 20 s, 52 °C for 20 s, and 72 °C for 45 s. All the samples were tested in triplicate.

### 4.8. Statistical Analysis

Statistical analyses were performed using IBM SPSS Statistic 29.0.0.0 (241) (New York, NY, USA), and graphical representation was performed using the SigmaPlot 14.5 software (SYSTAT, Palo Alto, CA, USA). Data were tested for normality with the Kolmogorov–Smirnov test. Since data distribution in all the samples significantly deviated from the normal distribution, the statistical significance of the observed differences was tested with non-parametric tests. Single comparisons were tested using the Mann–Whitney U test. Multiple comparisons were tested with Kruskal–Wallis H test, followed by pairwise analysis with Bonferroni correction. *p*-values were considered statistically significant if they were less than 0.05 */#, 0.01 **/##, or 0.001 ***/###, indicating the level of significance.

## Figures and Tables

**Figure 1 ijms-25-06179-f001:**
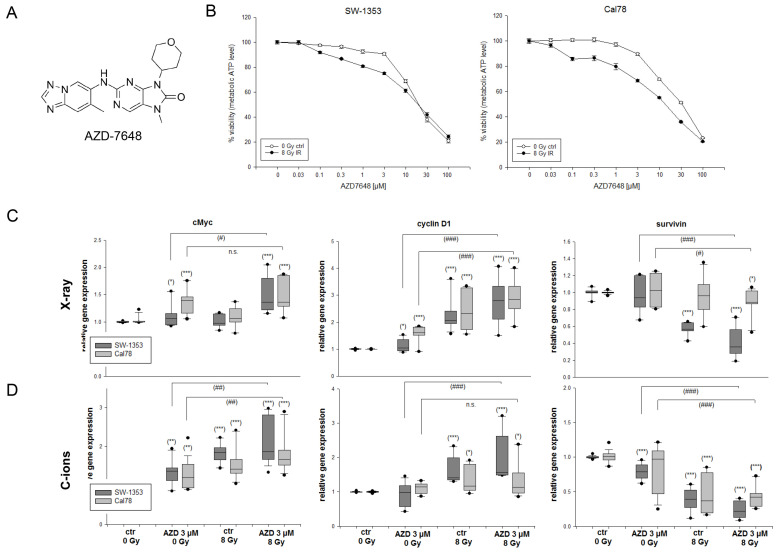
(**A**) Chemical structure of 7,9-dihydro-7-methyl-2-[(7-methyl[1,2,4]triazolo[1,5-a]pyridin-6-yl)amino]-9-(tetrahydro-2H-pyran-4-yl)-8H-purin-8-one (AZD7648). (**B**) The dose–response relationship after treatment with 0.03 and 100 µM AZD7648 with and without IR. The relative gene expression of the proto-oncogene cMyc, cyclin D1, and survivin, which play a role in cell cycle progression, apoptosis, and cellular transformation. The SW-1353 (dark grey) and Cal78 (light grey) chondrosarcoma cells were treated with 3 µM AZD7648 alone, respectively, in a combined treatment with (**C**) 8 Gy X-ray or (**D**) 8 Gy C-ion IR (mean ± SD, n = 6, measured in triplicates). Statistical significances to the untreated controls (ctrl 0 Gy) are defined as follows: * *p* < 0.05; ** *p* < 0.01; *** *p* < 0.001. Statistical significances between the 3 µM AZD7648 group and the combined treatment with IR are presented as # *p* < 0.05; ## *p* < 0.01; ### *p* < 0.001.

**Figure 2 ijms-25-06179-f002:**
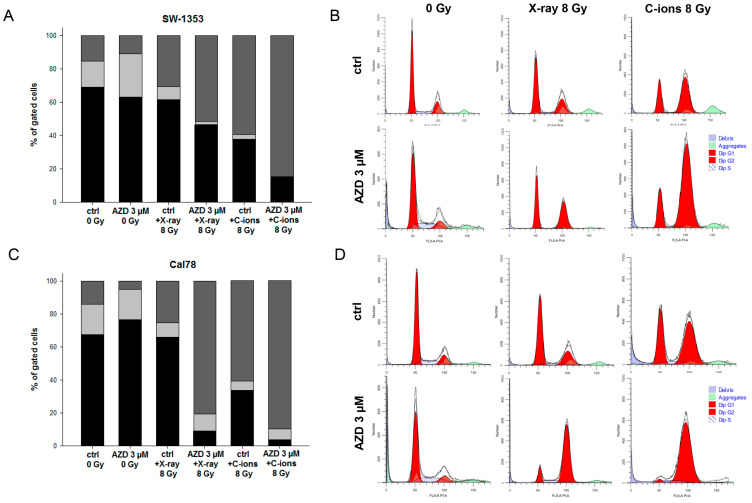
Cell cycle distribution after the combined treatment of the DNA-PKcs inhibitor AZD7648 and 8 Gy X-ray or C-ion IR. (**A**) The SW-1353 and (**C**) Cal78 chondrosarcoma cells were analyzed using flow cytometry 24 h after the combined treatment with 3 µM AZD7648 and 8 Gy X-ray or C-ion IR (LET ca. 55 keV/μm). The statistical evaluation is presented in stacked bar charts (n = 5). The distribution of cells in the G_1_/G_0_ phase (black), the S-phase (light gray), and the G_2_/M phase (dark gray) are shown in % of gated cells. (**B**,**D**) The representative original tracks of non-IR control cells (ctrl 0 Gy), 3 µM AZD7648 0 Gy (AZD 3 µM 0 Gy), and after 8 Gy X-ray, respectively, and C-ion IR with and without the AZD7648 treatment are shown. In both cell lines, C-ion IR induced a stronger G_2_/M arrest and a synergistic effect of the combined treatments.

**Figure 3 ijms-25-06179-f003:**
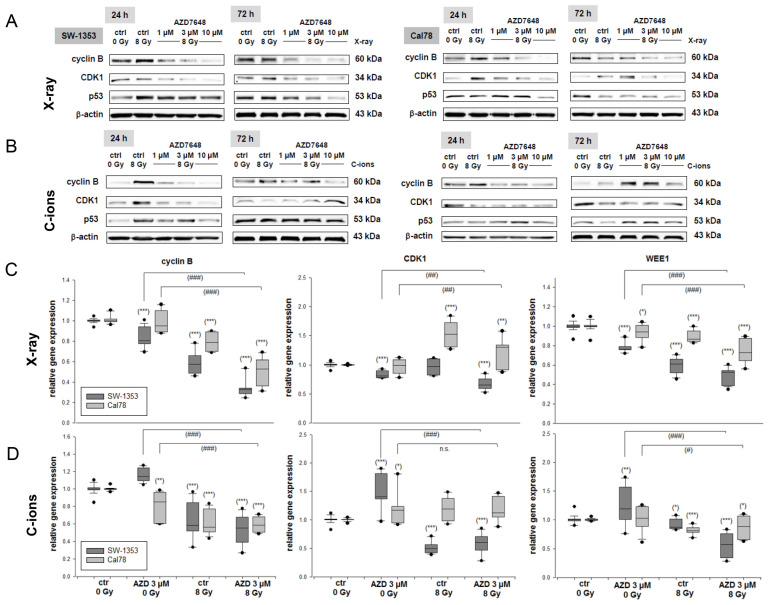
G_2_/M checkpoint genes regulation. The protein expression of the cell cycle G2/M phase key regulators cyclin B, cyclin-dependent kinase (CDK)1, and p53 was analyzed by immunoblotting in two chondrosarcoma cell lines under control conditions (ctrl 0 Gy), irradiated with (**A**) 8 Gy X-ray or (**B**) 8 Gy C-ion IR (ctrl 8 Gy), and after the combined treatment with 1 µM, 3 µM, and 10 µM AZD7648. β-actin was used as loading control (mean ± SD of n = 3). The full-length blots are presented in the Appendix A. The AZD7648 treatment sustainably reduced the expression of these cell cycle regulators, both 24 h and 72 h after IR. (**B**) The relative gene expression of the G_2_/M checkpoint genes cyclin B, CDK1, and WEE1 was significantly reduced by IR alone and the combined treatment. The SW-1353 (dark grey) and Cal78 (light grey) chondrosarcoma cells were treated with 3 µM AZD7648 alone, respectively, in a combined treatment with (**C**) 8 Gy X-ray or (**D**) 8 Gy C-ion IR (mean ± SD, n = 6, measured in triplicates). Statistical significances to the untreated controls (ctrl 0 Gy) are defined as follows: * *p* < 0.05; ** *p* < 0.01; *** *p* < 0.001. Statistical significances between the 3 µM AZD7648 group and the combined treatment with IR are presented as # *p* < 0.05; ## *p* < 0.01; ### *p* < 0.001.

**Figure 4 ijms-25-06179-f004:**
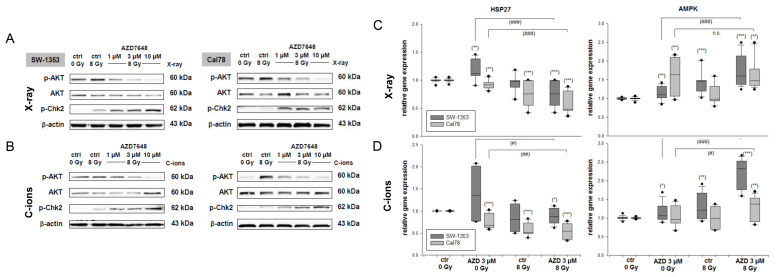
MAPK phosphorylation and the expression of protein stability markers. The p-AKT and p-Chk2 protein phosphorylation pattern of the SW-1353 and Cal78 chondrosarcoma cells 1 h after the combined treatment with 1 µM, 3 µM, or 10 µM AZD7648 and (**A**) 8 Gy X-ray, respectively (**B**) C-ion IR. While AKT phosphorylation was significantly downregulated, Chk2 phosphorylation was increased by the additional AZD7648 treatment. β-actin was used as loading control (mean ± SD of n = 3). The full-length blots are presented in the Appendix A. The relative gene expression of the protein stability marker HSP90 and (**D**) AMPK 24 h after combined treatment with 3 µM AZD7648 and (**C**) 8 Gy X-ray, respectively, (**D**) 8 Gy C-ion IR (SW-1353: dark grey; Cal78: light grey) (mean ± SD; n = 6; measured in triplicates). Non-IR cells (ctrl 0 Gy) were used as controls (ratio = 1). Statistical significances to the untreated controls (ctrl 0 Gy) are defined as follows: * *p* < 0.05; ** *p* < 0.01; *** *p* < 0.001. Statistical significances between the 3 µM AZD7648 group and the combined treatment with IR are presented as # *p* < 0.05; ## *p* < 0.01; ### *p* < 0.001.

**Figure 5 ijms-25-06179-f005:**
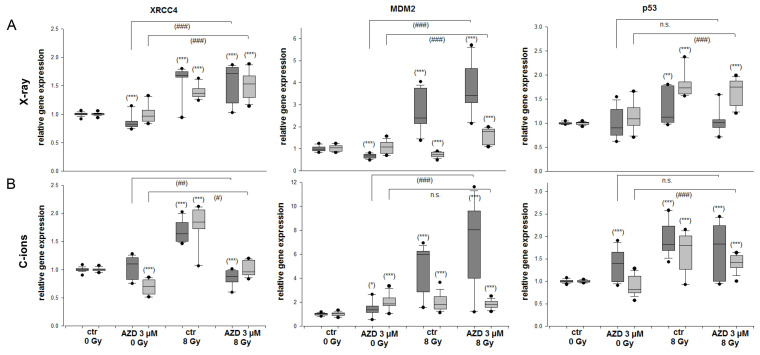
Relative gene expression of the DNA repair marker XRCC4 and die MDM2-p53 feedback loop. The SW-1353 (dark grey) and Cal78 (light grey) chondrosarcoma cells were treated with 3 µM AZD7648 alone, respectively, in a combined treatment with (**A**) 8 Gy X-ray or (**B**) 8 Gy C-ion IR (mean ± SD, n = 6, measured in triplicates). Statistical significances to the untreated controls are defined as follows: * *p* < 0.05; ** *p* < 0.01; *** *p* < 0.001. Statistical significances between the 3 µM AZD7648 group and the combined treatment with IR are presented as # *p* < 0.05; ## *p* < 0.01; ### *p* < 0.001.

**Figure 6 ijms-25-06179-f006:**
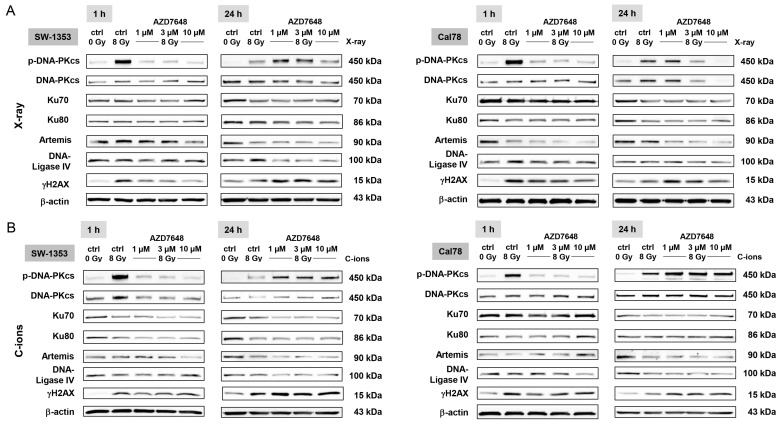
Changes in the NHEJ DNA repair pathway through DNA-PKcs inhibition. The protein expression of the NHEJ DNA repair pathway members Ku70, Ku80, Artemis, DNA Ligase IV, and the DNA damage marker γH2AX was analyzed by immunoblotting in two chondrosarcoma cell lines under control conditions (ctrl 0 Gy), irradiated with (**A**) 8 Gy X-ray or (**B**) 8 Gy C-ion IR (ctrl 8 Gy), and after combined treatment with 1 µM, 3 µM, and 10 µM AZD7648. β-actin was used as loading control (mean ± SD of n = 3). The full-length blots are presented in the Appendix A. The treatment with AZD7648 significantly decreased the expression of these DNA repair proteins and resulted in the preservation of yH2AX phosphorylation.

**Figure 7 ijms-25-06179-f007:**
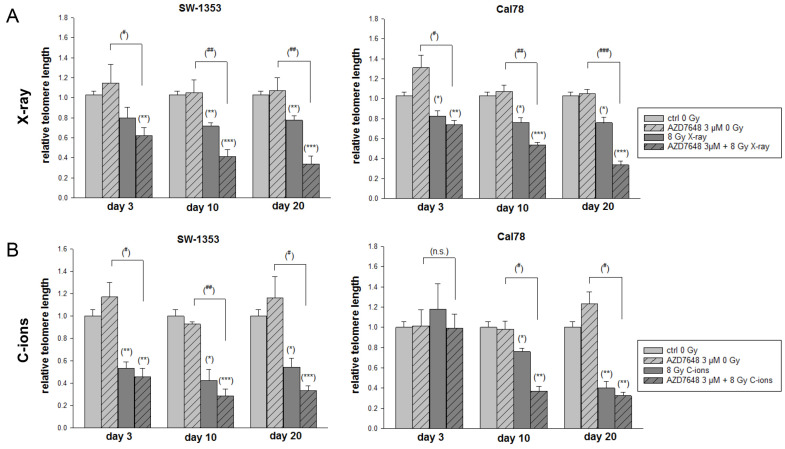
Telomere length reduction in the chondrosarcoma cells after combined treatment with AZD7648 and 8 Gy X-ray/C-ion IR. The SW-1353 and Cal78 cells were incubated with 3 µM AZD7648 (dashed light grey bars) and irradiated with (**A**) either 8 Gy X-ray or (**B**) 8 Gy C-ion IR (dark grey bars). After 3, 10, and 30 days, the DNA was isolated and the telomere length was determined. Statistical significances between the control group (ctrl 0 Gy) and the treated groups are defined as follows: * *p* < 0.05; ** *p* < 0.01; *** *p* < 0.001. Statistical significances between the 3 µM AZD7648 group (dashed light grey bars) and the combined treatment with IR (dashed dark grey bars) are presented as # *p* < 0.05; ## *p* < 0.01; ### *p* < 0.001.

**Table 1 ijms-25-06179-t001:** Cell cycle distribution of the chondrosarcoma cell lines 48 h after the combined treatment of 3 µM AZD7648 and 8 Gy X-ray, respectively, and C-ion IR (n = 5; mean ± SD; n.s.: not significant). Statistical significances to the non-irradiated controls (ctrl 0 Gy) are defined as follows: * *p* < 0.05; ** *p* < 0.01; *** *p* < 0.001. Statistical significances between irradiated controls (X-ray 8 Gy, respectively, C-ions 8 Gy) to the combined treatment with 3 µM AZD7648 are presented as # *p* < 0.05; ## *p* < 0.01; ### *p* < 0.001.

	SW-1353			Cal78		
	G_1_/G_0_	S	G_2_/M	G_1_/G_0_	S	G_2_/M
ctrl 0 Gy	69.2 ± 1.9	15.6 ± 2.5	15.2 ± 4.5	67.6 ± 0.4	18.3 ± 1.6	14.1 ± 2.0
3 µM AZD 0 Gy	63.2 ± 3.4 n.s.	25.9 ± 1.9 n.s.	10.8 ± 3.0 n.s.	76.6 ± 4.5 *	18.5 ± 2.5 n.s.	4.9 ± 2.5 *
X-ray 8 Gy	61.6 ± 0.8 *	7.8 ± 1.8 *	30.7 ± 2.5 *	65.8 ± 8.0 n.s.	8.9 ± 0.9 **	25.3 ± 8.8 n.s.
AZD 3 µM + X-ray 8 Gy	46.6 ± 2.1 ***, ##	1.9 ± 0.0 *, #	51.4 ± 1.6 ***, ###	9.1 ± 1.8 ***, ##	10.3 ± 0.4 *, #	80.5 ± 0.4 ***, ###
C-ions 8 Gy	37.8 ± 0.3 ***	2.8 ± 0.8 ***	59.4 ± 0.9 ***	33.8 ± 1.7 ***	5.4 ± 2.4 ***	60.9 ± 3.8 ***
AZD 3 µM + C-ions 8 Gy	15.1 ± 2.7 ***, ##	0.5 ± 0.2 ***	84.6 ± 2.3 ***, ###	3.8 ± 1.8 ***, ###	6.4 ± 1.6 ***	90.1 ± 1.2 ***, ###

## Data Availability

All data generated or analyzed during this study are included in this.

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
