# Peer review of "DNA-PKcs Inhibition Sensitizes Human Chondrosarcoma Cells to Carbon Ion Irradiation via Cell Cycle Arrest and Telomere Capping Disruption"

_ijms, 2024, doi:10.3390/ijms25116179_

Round 1
Reviewer 1 Report
Comments and Suggestions for Authors
Authors described that inhibition of DNA-PKcs may increase sensitivity to X-rays and C-ion irradiation by impairing its functional role in DNA repair mechanisms and telomere end protection. Although I found this article interesting, there are major issues with the explanation and/or interpretation of the data.
1. Line 60: please add a brief explanation of ATMiVE-821.
2. Line 92: "AZD7648 ± 8 Gy X-rays or C-ion IR" does not represent the conditions correctly. This is a misleading expression, so please explain it correctly.
3. In this paper, all gene expression analysis data are expressed as boxplots. However, all explanations of the results are expressed as mean ± standard deviation, which is very difficult to understand. Add a plot showing the mean value to the box plot or change it to a bar chart. Also, please explain in the figure legend what the black plot on the graph indicates.
4. Line 120: Please add an explanation for each color in the stacked bar graph.
5. Line 125: An additive AZD7648 treatment significantly decreased the number of gated cells in the G0/G1 phase.
6. Lines 149, 181, 228: To help the reader's understanding, please quantify the signal that indicates the protein band of interest.
7. Line 151: Authors stated "well correlated with the cell cycle arrest at G2/M". Please present the data objectively.
8. Line 151: The increased p53 expression after IR was reduced again by AZD7648 treatment. Please present the data objectively.
9. Line 154-155: This explanation is necessary from Fig.1.
10. Line 155-156: ”Both IR and the combined treatment led to a significant decrease in the expression of the G2/M checkpoint key genes”. Genes whose expression increases with both radiation and inhibitor use can be confirmed. Please state the results correctly. Furthermore, cyclinB1 protein increases in SW-1353 cells 24 hours after carbon ion irradiation, but gene expression decreases.
11. Fig.5: Authors forgot to add a legend to the graph.
12. Line 252: "IR and in particular the combined treatment with AZD7648 gradually decreased telomere length time dependently." Please perform statistical processing on the target data and show whether the decrease is time-dependent.
13. Can you provide basic data on cell proliferation and colony formation of cell lines when exposed to radiation or when inhibitors are added? This is because lines 236 and 254 contain statements regarding radiation sensitivity, so I think it is necessary to show the basic data for this experimental system.
Reviewer 2 Report
Comments and Suggestions for Authors
Dear Authors!
I have read your manuscript very carefully Aaron found it very interesting, well-written and well-organized. Your manuscript seeks to emerge DNA-PKcs as a therapeutic target on chondrosarcomas developing thus new radio-sensitizers. Although, your work is really important, I have a few concerns/suggestions before its publication.
1) I suggest to make more clear in your manuscript the aim of your study. You should refer it both in abstract and/or discussion section.
2) I could not understand how you chose the 8 Gy treatment and 1, 3 and 10μM treatment of AZD. Have you had any MTT or SRB assay finding the EC50 value? I strongly suggest to demonstrate in your manuscript those data if you have it. If you do not, you could perform MTT or SRB assay on both cell lines in order to find the EC50 value. Therefore, you can explain to the readers why you chose that treatment conditions.
Comments on the Quality of English LanguageEnglish language is fine.
Round 2
Reviewer 1 Report
Comments and Suggestions for Authors
The authors sincerely responded reviewer's comments and suggestion.
Author Response
The authors sincerely responded reviewer's comments and suggestion.
Authors reply: Thank you very much for your contribution to improving our manuscript.
Reviewer 2 Report
Comments and Suggestions for Authors
Dear Authors!
Thank you very much for taking into consideration my comments.
I think the addition of Figure 1B provides the appropriate information about
your experimental setup. Furthermore, it is more clear now the aim of your study that is presented in the discussion section.
I suggest your manuscript to be published, but you need first to decrease the similarity index from 29% to below 20%.
Author Response
Dear Authors!
Thank you very much for taking into consideration my comments.
I think the addition of Figure 1B provides the appropriate information about your experimental setup. Furthermore, it is more clear now the aim of your study that is presented in the discussion section.
I suggest your manuscript to be published, but you need first to decrease the similarity index from 29% to below 20%.
Authors reply: Thank you very much for your contribution to improving our manuscript.
According the plagiarism check report from the publisher we have reduced the duplication as much as we can (except the methods section - here we were guaranteed by the publisher that this is excluded). Many of these similarities arise from the large number of publications by our working group in the radio-oncology of malignant bone tumours and are very difficult to avoid. We hope that the changes have been made to your satisfaction.